# Germline Variation in *PDCD1* Is Associated with Overall Survival in Patients with Metastatic Melanoma Treated with Anti-PD-1 Monotherapy

**DOI:** 10.3390/cancers13061370

**Published:** 2021-03-18

**Authors:** Mirjam de With, Daan P. Hurkmans, Esther Oomen-de Hoop, Ayoub Lalouti, Sander Bins, Samira El Bouazzaoui, Mandy van Brakel, Reno Debets, Joachim G. J. V. Aerts, Ron H. N. van Schaik, Ron H. J. Mathijssen, Astrid A. M. van der Veldt

**Affiliations:** 1Department of Medical Oncology, Erasmus MC Cancer Institute, 3015 GD Rotterdam, The Netherlands; m.dewith@erasmusmc.nl (M.d.W.); e.oomen-dehoop@erasmusmc.nl (E.O.-d.H.); 424383al@student.eur.nl (A.L.); s.bins@erasmusmc.nl (S.B.); m.vanbrakel@erasmusmc.nl (M.v.B.); j.debets@erasmusmc.nl (R.D.); a.mathijssen@erasmusmc.nl (R.H.J.M.); a.vanderveldt@erasmusmc.nl (A.A.M.v.d.V.); 2Department of Clinical Chemistry, Erasmus MC Cancer Institute, 3015 GD Rotterdam, The Netherlands; s.elbouazzaoui@erasmusmc.nl (S.E.B.); r.vanschaik@erasmusmc.nl (R.H.N.v.S.); 3Department of Pulmonology, Erasmus MC Cancer Institute, 3015 GD Rotterdam, The Netherlands; j.aerts@erasmusmc.nl; 4Department of Radiology & Nuclear Medicine, Erasmus MC Cancer Institute, 3015 GD Rotterdam, The Netherlands

**Keywords:** immune checkpoint inhibitors, PD-1, metastatic melanoma, germline variation, autoimmunity

## Abstract

**Simple Summary:**

Although the introduction of programmed cell death 1 (PD-1) immune checkpoint inhibitors (ICIs) has significantly improved the overall survival (OS) in patients with metastatic melanoma, a substantial number of patients do not benefit from ICIs. Therefore, the predictive value of single nucleotide polymorphisms (SNPs) in genes related to the PD-1 axis was investigated in patients with metastatic melanoma and anti-PD-1 monotherapy. Germline variation in the gene encoding for PD-1, *PDCD1* 804C > T (rs2227981), was associated with poorer OS with a 3-year OS rate of 51.8%, as compared to 71% in wild type patients. In addition, *PDCD1* 804C > T carriers had significantly lower mRNA expression in several tissues and a decreased fraction of PD-1^+^ CD4^+^ T cells, indicating that *PDCD1* 804C > T may affect clinical benefit from ICIs by decreasing transcriptional initiation and PD-1 expression in T cells. These findings show that germline genetics may significantly impact immune responses after ICIs.

**Abstract:**

A substantial number of melanoma patients do not benefit from therapy with anti-PD-1. Therefore, we investigated the predictive value of single nucleotide polymorphisms (SNPs) in genes related to the PD-1 axis in patients with metastatic melanoma. From 119 consecutive melanoma patients who were treated with pembrolizumab or nivolumab monotherapy, blood samples were genotyped for 11 SNPs in nine genes. Associations between SNPs and OS were tested using Cox regression analysis and internally validated by bootstrapping. For SNPs with a statistical significance, an expression quantitative trait loci (eQTL) analysis was performed. In a subset of patients, immunophenotyping was performed. Patients with a SNP in *PDCD1* (804C > T; rs2227981) had a significantly poorer OS with a 3-year OS rate of 51.8%, as compared to 71% in wild type patients (hazard ratio [HR] 2.37; 95% CI: 1.11–5.04; *p* = 0.026). eQTL analysis showed that this SNP was associated with decreased gene expression. In addition, *PDCD1* 804C > T carriers had a reduced fraction of peripheral PD-1^+^CD4^+^ T cells. No other associations between SNPs and OS were found. *PDCD1* 804C > T is associated with poorer OS after anti-PD-1 monotherapy in patients with metastatic melanoma. This SNP may affect clinical benefit from ICIs by decreasing transcription initiation and expression of PD-1 in T cells.

## 1. Introduction

Melanoma is the most aggressive skin cancer, with a poor prognosis in patients with advanced disease [1]. More than 100,000 new cases occur in Europe each year, and the incidence continues to increase [1]. Spontaneous regression of melanoma is observed in about 15% of patients with primary melanoma [2]. In addition, spontaneous regression of primary melanoma may occur in advanced stage disease, as primary melanoma cannot be detected in approximately 5% of patients with metastatic melanoma [2]. This spontaneous regression is the result of antitumor immunity with lymphocytic and histiocytic infiltration in primary melanomas, which may eventually result in the disappearance of tumor cells or even normal bystander melanocytes [3,4].

Vitiligo is a common autoimmune disease with depigmentation of the skin that is caused by the destruction of melanocytes by a T cell-mediated autoimmune response. In patients with metastatic melanoma, vitiligo is associated with improved overall survival (OS) [5]. The combination of antitumor immunity (i.e., spontaneous regression) and autoimmunity (i.e., vitiligo) in patients with melanoma is considered an effective immune response against melanocytic cells [6].

As effective spontaneous antitumor immunity is rare, immune checkpoint inhibitors (ICIs) have been developed to reinforce the endogenous immunity. Cancer cells are known to impair antitumor immunity by affecting the cytotoxic T-lymphocyte-associated protein 4 (CTLA-4) and programmed cell death 1 (PD-1) axes with the subsequent downregulation of the T cell effector function [7]. While the introduction of ICIs targeting CTLA-4 was the first step towards improvement of OS in patients with advanced melanoma, PD-1 inhibitors further improved 5-year survival up to 52% [8]. Comparable to observations in melanoma patients with spontaneous remission, immune-related adverse events (irAEs) are associated with prolonged OS in patients with advanced melanoma who are treated with ICIs [9].

Therefore, it is conceivable that factors which are associated with autoimmunity may predict benefit from ICIs. For several germline variants, or single nucleotide polymorphisms (SNPs), associations with different autoimmune diseases have been described. SNPs contribute to the 0.1% difference in the DNA sequence of the human genome [10]. Nonsynonymous SNPs may alter the amino acid sequence; while both synonymous and nonsynonymous SNPs can influence gene expression, messenger RNA stability, and translational efficiency [10,11]. Thereby altering activity, function or quantity of the encoded proteins. Although some SNPs are harmless, other SNPs can cause diseases. For example, an increased risk of autoimmune vitiligo is associated with a germline variant in *HLAA*, a gene encoding for the major histocompatibility complex (MHC) that binds to the T cell receptor and stimulates T cell activation [12].

As autoimmunity may reflect antitumor immunity, the primary objective of the current study was to investigate whether SNPs in genes related to the PD-1 axis and involved in autoimmunity are predictive for OS in patients with metastatic melanoma and anti-PD-1 monotherapy. The secondary objectives were to assess the association of those germline variants with progression-free survival (PFS) and best overall response (BOR). To better understand the results, translational analyses were also performed, including multi-tissue expression quantitative trait loci (eQTL) analysis and peripheral immunophenotyping by flow cytometry.

## 2. Materials and Methods

### 2.1. Study Design

In this single-center study at Erasmus University Medical Center (Rotterdam, The Netherlands), prospectively collected whole blood samples were analyzed to determine germline variations in patients with metastatic melanoma who were treated with anti-PD-1 monotherapy. Patients were included if they fulfilled the following inclusion criteria: anti-PD-1 monotherapy (i.e., pembrolizumab or nivolumab), a minimum follow-up of one year at data cut-off, availability of a whole blood sample, and signed informed consent for DNA analysis (Erasmus Medical Center ethics board, Rotterdam, The Netherlands, study number MEC 02-1002). A subgroup of patients signed an additional informed consent for immunomonitoring (MULTOMAB study; Erasmus Medical Center ethics board, Rotterdam, The Netherlands, study number MEC 16-011).

### 2.2. Collection of Clinical Data

At the initiation of monotherapy with anti-PD-1, the following patient and disease characteristics were collected: age, gender, *BRAF* V600E/K mutation status, prior systemic therapy, prior radiotherapy, Eastern Cooperative Oncology Group (ECOG) performance status, serum lactate dehydrogenase (LDH; U/L), and presence of central nervous system (CNS) metastases. The cut-off for follow-up data was set at 9 December 2019.

OS was defined from the start of treatment to death by any cause. Patients were censored at the date on which they were last known to be alive. PFS was defined as the time from the start of treatment to the first documented progressive disease (PD) according to Response Evaluation Criteria in Solid Tumours version 1.1 (RECIST v1.1) [13] or death by any cause. The therapy response was determined according to RECIST v1.1, i.e., complete response (CR), partial response (PR), stable disease (SD), or PD. For BOR, confirmation of PR and CR was not required, and a minimum duration of 90 days was required for SD.

### 2.3. Selection of SNPs

SNPs in genes related to the PD-1 axis were selected based on their previous reported associations with autoimmunity in the literature. On 15 June 2018, a literature search was performed for the combination of polymorphisms, autoimmunity and genes that were related to T cell receptor (TCR) signaling (Kyoto Encyclopedia of Genes and Genomes [KEGG] TCR pathway) or the PD-1 pathway according to the QIAGEN Ingenuity Pathway Analysis (QIAGEN, Redwood City, CA, USA). SNPs were prioritized according to their strong associations with susceptibility to autoimmune diseases (generally reflected by the odds ratio (OR)) or their validation in independent populations. Common SNPs with a minor allele frequency (MAF) above 5% were included. Synonymous SNPs were excluded, except for SNPs with a predicted variant effect (i.e., intronic variation in the promotor region affecting mRNA transcription; estimated by the variant effect predictor tool; https://ensembl.org, accessed on 24 September 2018). Finally, eleven separate SNPs in nine genes were selected for the current analyses (Table 1). An overview of the search strategy is depicted in Appendix A.

### 2.4. DNA Isolation and Genotyping

DNA was isolated from 400 μL of the whole blood specimens on the MagNaPure Compact Instrument (Roche Diagnostics GmbH, Mannheim, Germany) using the Nucleic Acid Isolation Kit I (Roche Diagnostics GmbH, Mannheim, Germany). Predesigned drug metabolism enzymes (DME) TaqMan allelic discrimination assays (Table 2) were used to genotype the selected SNPs on the Life Technologies TaqMan 7500 system (Applied Biosystems, Life Technologies Europe BV, Bleiswijk, The Netherlands). Each assay consisted of two allele-specific minor groove binding probes (MGB) labeled with 2′-chloro-7′phenyl-1,4-dichloro-6-carboxy-fluorescein (VIC) and 6-caboxyfluorescein (FAM) fluorescent dyes. To conduct qPCR, 20 ng of genomic DNA was analyzed using the assays and a TaqMan GTX press Master Mix (Applied Biosystems, Life Technologies Europe BV, Bleiswijk, The Netherlands). qPCR consisted of 40 cycles of denaturation (95 °C for 20 s), followed by annealing (92 °C for 3 s) and extension (60 °C for 30 s). Using the TaqMan 7500 software v2.3 for allelic discrimination (Applied Biosystems, Life Technologies Europe BV, Bleiswijk, The Netherlands), the genotypes were determined by measuring the allele-specific fluorescence.

### 2.5. Multi-Tissue Expression Quantitative Trait Loci (eQTL) Analysis

For germline variations with statistical significance in multivariate analyses, expression quantitative trait loci (eQTL) analysis was performed to determine the effect of genetic variations on gene expression in normal tissue. Global RNA expression was analyzed for the minor and major allele of the germline variant using the Genotype-Tissue Expression (GTEx) database (https://gtexportal.org, accessed on 17 May 2020) [23]. This open-source GTEx project allows one to study variation in gene expression levels of diverse tissues of the human body. The minor and major allele were tested against gene expression using linear regression analysis. The normalized effect size (NES) is defined as the slope of the linear regression and computed as the effect of the minor allele.

### 2.6. Flow Cytometry

From patients who signed an additional informed consent, peripheral blood mononuclear cells (PBMCs) were isolated at baseline (prior to the first administration) and during anti-PD-1 monotherapy (prior to the second and third administration). As previously described [24], flow cytometry was performed and frequencies of CD4^+^ and CD8^+^ T cells expressing PD-1 were determined. In addition, frequencies of regulatory T cells (Tregs) expressing PD-1 were measured.

### 2.7. Statistical Analysis

Data were presented as the prevalence (percentage) or median (range). The distribution of the genotypes was tested for Hardy–Weinberg equilibrium (HWE) using the Chi-square test. Two genetic models (i.e., the dominant and recessive model) were applied for further analyses [25]. To study associations between clinical baseline characteristics or SNPs with survival (OS and PFS), Cox proportional hazards regression analysis was used to calculate the HR with 95% confidence intervals (CIs). Variables with *p* ≤ 0.10 in univariable analyses were included in multivariable analyses where backward selection was applied with a threshold of *p* < 0.05. The proportional hazards assumption was determined for all Cox analyses by means of the Schoenfeld residuals. For SNPs with *p* < 0.10 in univariate analyses, Kaplan–Meier curves were applied for PFS and OS.

To identify patients with clinical benefit according to BOR, patients were categorized into “clinical benefit” (i.e., CR, PR, and SD) and “no clinical benefit” (i.e., PD). The genetic models were tested against BOR using logistic regression. The odds ratio (OR) and 95% CIs were calculated.

All associations with *p* < 0.10 were internally validated by bootstrapping [26]. For this purpose, one thousand bootstrap samples were generated (with replacements) and bias-corrected 95% CIs were calculated for the ORs and HRs.

The Mann–Whitney U test was used to test whether frequencies of CD4^+^ and CD8^+^ T cells with PD-1 expression were different between wild types and homozygous + heterozygous variants.

Correlation analyses of clinical baseline characteristics and SNPs were performed by either the rank-biserial (for ordinal variables) or Phi correlation test (for continuous variables) using IBM SPSS Statistics (version 24.0.0.1, Chicago, IL, USA). All other statistical analyses were performed using STATA (version 15.1, StataCorp LP, Lakeway Drive College Station, TX, USA).

## 3. Results

### 3.1. Clinical Characteristics

In total, 119 patients with metastatic melanoma and anti-PD-1 monotherapy were included. Baseline characteristics are shown in Table 3. Sixty-three patients (53%) were treated with twice-weekly nivolumab, (weight-based 3 mg/kg) and 56 patients (47%) were treated with thrice-weekly pembrolizumab (weight-based 2 mg/kg). Prior to the approval of anti-PD-1, one patient had been treated with pembrolizumab in the Keynote-001 trial (NCT01295827) [27], and one patient had been treated with pembrolizumab in a compassionate use program. At baseline, 32 patients (27%) had an ECOG performance status ≥1 and the median level of serum LDH was 211 U/L (range 118–1523 U/L). In the majority of patients (*N* = 89), anti-PD-1 monotherapy was administered as first-line treatment, whereas 30 patients (25%) had received at least one prior treatment line, including BRAF-MEK-inhibitors (*N* = 14), ipilimumab (*N* = 7), a sequential combination of both (*N* = 4), or talimogene laherparepvec (TVEC) (*N* = 5).

The median follow-up was 2.6 years (range 0.2–4.2 years). At data cut-off, 75 patients (63%) were still alive. The median OS was not reached and the median PFS was 19.6 months (range 9.0–30.1 months). In 117 patients, response evaluation was available for further analyses. Two patients were excluded for response evaluation due to insufficient baseline scans for radiological evaluation. After the initiation of anti-PD-1 monotherapy, 86 (73.5%) patients had clinical benefit, with 26 (22%), 43 (37%), and 17 (15%) patients having a CR, PR, and SD, respectively. Thirty-one patients (26%) had PD. During follow-up, 67 out of 117 patients (57%) eventually had PD according RECIST v.1.1.

### 3.2. Germline Variation Associated with OS, PFS and BOR

All investigated SNPs were in HWE (Table 2).

Among clinical baseline characteristics, age (≥65 vs. <65 years), ECOG performance status (≥1 vs. 0), and LDH levels (per U/L) were significantly associated with OS (HR 1.695, 95%CI 0.910–3.94; HR 3.625, 95% CI 2.037–6.651; and HR 1.004, 95%CI 1.000–1.005, respectively). In univariable analyses, germline variation in *GZMB* and *PDCD1* showed a trend towards associations with OS according to the dominant model (HR 1.811, 95%CI 0.945–3.364, *p* = 0.060 and HR 2.027, 95%CI 0.998–4.118, *p* = 0.051, respectively). After correction for baseline factors, patients with the germline variant *PDCD1* 804C > T still had a significantly worse OS with a 3-year survival rate of 51.8%, as compared to 71.0% in wild type patients (OR 2.366; 95% CI 1.111–5.036; *p* = 0.026). These results were confirmed by internal validation (bias corrected 95%CI 1.039–6.246). As *GZMB* c.128C > A was not significant in the multivariable analysis, the SNP was excluded from the multivariable model. No violations of the proportional hazards assumption for Cox analyses were found. Univariable and multivariable analyses for OS are shown in Table 4. Appendix A show the OS curves of patients with metastatic melanoma according to germline variation in PDCD1 (804C > T) and GZMB (c.128C > A), respectively.

Based on univariable analysis for PFS (*p* < 0.10), the following clinical baseline characteristics could be included in multivariate PFS analysis: ECOG performance (≥1 vs. 0), prior treatment (yes vs. no), *BRAF* mutation status, and LDH level (per U/L). In univariable analysis for PFS, *GZMB* c.128C > A fulfilled criteria for multivariable analyses according to the dominant model (HR 1.633, 95%CI 0.999–2.670, *p* = 0.051; Appendix A). After correction for clinical baseline characteristics, germline variation *GZMB* c.128C > A was not associated with PFS (*p* = 0.159). No violations of the proportional hazards assumption for Cox analyses were found. Appendix A show the PFS curves according to germline variation in PDCD1 (804C > T) and GZMB (c.128C > A), respectively.

There were no significant associations between the SNPS and BOR (Appendix A).

Clinical baseline characteristics and SNPs with *p* < 0.10 were tested for their correlation. No significant correlations between SNPs and clinical baseline characteristics were found. Prior treatment was significantly correlated with poorer ECOG performance status (correlation coefficient 0.203, *p* = 0.030) and a *BRAF* mutation was correlated with age (correlation coefficient -0.340, *p* < 0.001) (Appendix A).

### 3.3. PDCD1 804C > T and mRNA Expression in Human Tissue

In multivariable analysis, germline variation in *PDCD1* (804C > T) remained significantly associated with OS. Moreover, the upstream variant *PDCD1* 804C > T was estimated to have regulatory implications (by the variant effect predictor tool). Next, an eQTL analysis was conducted to determine the impact of this SNP on mRNA expression of *PDCD1* across tissue. This analysis showed that germline variation in *PDCD1* (804C > T) is significantly associated with lower gene expression of *PDCD1* in human tissue, in particular in subcutaneous adipose tissue (Appendix A).

### 3.4. PDCD1 804C > T and PD-1 Expression in Circulating T Cells

In 47 patients, the impact of germline variation *PDCD1* 804C > T on the PD-1 expression in circulating immune cell populations could be investigated in blood obtained prior to and after anti-PD-1 monotherapy. There was a consistent tendency that patients with heterozygous or homozygous variation of *PDCD1* 804C > T had a lower fraction of PD-1^+^ CD4^+^ T cells, which was observed in PBMCs prior to and during anti-PD-1 monotherapy (Figure 1A–C). In contrast, this tendency was not observed in CD8^+^ T cells and regulatory T cells (Figure 1D–F and Appendix A, respectively).

## 4. Discussion

The current study demonstrates that patients with metastatic melanoma and a SNP in *PDCD1* (804C > T; rs2227981) have a significantly poorer OS after treatment with anti-PD-1 monotherapy, resulting in a 3-year survival rate of 51.8%, as compared to 71.0% for wild type patients. These results were internally validated and corrected for prognostic factors, including ECOG performance status and LDH. In addition, the SNP was associated with the decreased gene expression of *PDCD1* in several tissues, in particular in subcutaneous adipose tissue. Besides, there was a consistent tendency that patients with heterozygous or homozygous variation of *PDCD1* 804C > T had a lower fraction of PD-1^+^ CD4^+^ T cells.

The PD-1 receptor is encoded by *PDCD1* and is primarily expressed on CD4^+^ and CD8^+^ T cells [28]. The interaction of PD-1 on T cells to its ligands 1 or 2 (PD-L1 or PD-L2) suppresses T cell proliferation and function. As the SNP in *PDCD1* (804C > T) is located in the promoter region of the gene, it may affect transcriptional initiation, resulting in a decreased expression of PD-1. In the current study, we have demonstrated this decreased PD-1 expression in both tissue and peripheral blood, thereby explaining the poorer OS in patients with the *PDCD1* 804C > T SNP. First, the locus was found to control *PDCD1* transcription in tissues and the SNP (804C > T) was predicted to negatively impact transcription. Second, eQTL analysis showed that the SNP in *PDCD1* (804C > T) was associated with a significantly lower mRNA expression in several human tissues, in particular in subcutaneous adipose tissue. This strong relation between a SNP in *PDCD1* (804C > T) and *PDCD1* expression in subcutaneous adipose tissue may be explained by the high fraction of CD4^+^ T cells with PD-1 expression in subcutaneous adipose tissue [29]. Third, peripheral immunophenotyping showed a trend towards a decreased fraction of PD-1^+^ CD4^+^ T cells in patients with the *PDCD1* 804C > T SNP. Likewise, patients with type 1 diabetes and the *PDCD1* 804C > T SNP have a significantly lower fraction of PD-1 expression in peripheral CD4^+^ T cells [30], which supports the observed trend. Together, these three findings suggest that the *PDCD1* 804C > T SNP affects transcriptional initiation and consequently results in a lower expression of PD-1 in CD4^+^ T cells.

In addition, patients with the *PDCD1* 804C > T SNP had an almost 20% lower 3-year survival rate as compared to wild type patients. This may be caused by a decreased PD-1 expression in T cells. Previously, PD-1 expression in tumor infiltrating lymphocytes has been associated with survival in several tumor types [31,32]. For example, patients with positive PD-1 immunostaining in melanoma lymph node metastases have improved melanoma-specific survival compared to patients without PD-1 expression [33].

Although *PDCD1* 804C > T was associated with poorer OS after anti-PD-1 monotherapy in our study, we did not observe significant associations between *PDCD1* 804C > T and PFS or BOR. However, ICI-induced tumor response is not always predictive of OS, as improved OS has been reported in patients with metastatic melanoma after continuation of anti-PD-1 beyond disease progression [34]. Besides, the use of RECIST v1.1 in this study may explain the discrepancy. While RECIST v1.1 is still used in routine clinical practice, tumor response to anti-PD-1 therapy can be underestimated in 15% of patients [35].

As observed previously in patients with non-small cell lung cancer (NSCLC) [36], patients with heterozygous or homozygous variation of *GZMB* c.128C > A have significantly worse PFS and OS after anti-PD-1 monotherapy in univariable analysis. After correction for baseline characteristics, however, these associations did not retain significance. *GZMB* encodes for the similar named protease granzyme B, that is released by effector T cells to induce apoptosis of tumor cells [37], and the c.128C > A SNP is hypothesized to define an isoform of granzyme B that is incapable of apoptosis in tumor cell lines [38]. As we found no significant association in this cohort between *GZMB* c.128C>A and survival in multivariable analysis, the clinical relevance of this SNP seems limited and needs to be verified in larger cohorts.

As SNPs in genes related to the PD-1 axis are associated with autoimmune diseases [12,14,15,16,17,18,19,20,21,22] and irAEs are correlated with treatment outcome of ICIs [39,40], it is conceivable that these SNPs are also associated with irAEs. In the current study, we did not evaluate irAEs, as the retrospective design would have resulted in underreporting of irAEs, in particular of low grade irAEs such as vitiligo. Although a previous retrospective study in patients with NSCLC did not show any association between *PDCD1* 804C > T and irAEs [41], prospective studies are needed to investigate the role of autoimmune disease-related SNPs in the development of irAEs.

Due to the retrospective data collection in this study, brain scans were not available for all patients at baseline. As a result, multivariable analyses could not be corrected for CNS metastases, a well-known prognostic factor in metastatic melanoma. Other limitations of the current study include the limited sample size and the lack of an external validation cohort, which was overcome by internal validation. Nevertheless, the large difference in 3-year OS rates between carriers and non-carriers of the 804C > T SNP in *PDCD1* indicates that germline mutations in genes related to the PD-1 axis may have a significant impact on the outcome of patients with cancer after ICIs. Genome-wide association studies (GWAS) may contribute to the further identification of germline variants, affecting the delicate balance between autoimmunity and antitumor immunity after ICIs.

## 5. Conclusions

The present study demonstrates that a common upstream germline variant of *PDCD1* (804C > T; rs2227981) is associated with worse OS in patients with metastatic melanoma after treatment with PD-1 ICIs. This SNP may affect clinical benefit from ICIs by decreasing transcription initiation and expression of PD-1 in T cells. These observations highlight the clinical need to understand how germline genetics affects immune responses during immunotherapy, and suggest the feasibility for future GWAS in immuno-oncology.

## Figures and Tables

**Figure 1 cancers-13-01370-f001:**
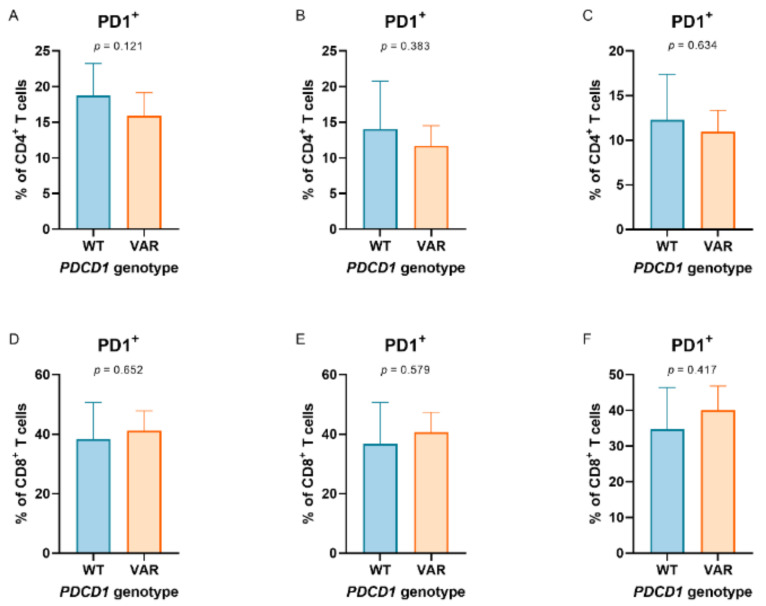
Programmed cell death 1 (PD-1) expression in peripheral T cell populations according to PDCD1 804C > T genotypes. PD-1 expression in peripheral (**A**–**C**) CD4+ and (**D**–**F**) CD8+ T cells according to PDCD1 804C > T genotypes (comparing wild types (WT) and heterozygous variants + homozygous variants (VAR)). Peripheral T cells were collected prior to (baseline, graphs on the left) and during anti-PD-1 monotherapy (mid: prior to the second administration of anti-PD-1 monotherapy; right: prior to the third administration of anti-PD-1 monotherapy) in patients with metastatic melanoma (*N* = 47).

**Table 1 cancers-13-01370-t001:** Literature overview of selected single nucleotide polymorphisms (SNPs) and their associations with autoimmunity.

Gene	SNP ID	Association	OR (95% CI)	*p* Value	Reference
*GZMB*	rs8192917	Vitiligo	1.25 (1.18–1.32)	8.91 × 10^−16^	Jin et al. [12]
*HLAA*	rs60131261	Vitiligo	1.54 (1.46–1.61)	1.56 × 10^−66^	Jin et al. [12]
*IFNG*	rs2430561	Idiopathic thrombocytopenic purpura	1.75 (1.23–2.50)	0.002	Lee et al. [14]
		Systemic lupus erythematosus	2.48 (1.14–5.41)	0.022	Kim et al. [15]
	rs2069705	Systemic lupus erythematosus	2.27 (1.34–3.87)	0.0024	Kim et al. [15]
	rs2069718	Systemic lupus erythematosus	2.42 (1.11–5.27)	0.026	Kim et al. [15]
*IL10*	rs3024493	Crohn’s disease, ulcerative colitis, type 1 Diabetes, Behçet’s disease	1.26 (1.13–1.41)	4.38 × 10^−5^	Ramos et al. [16]
*IL2RA*	rs2104286	Rheumatoid arthritis	0.92 (0.87–0.97)	0.001	Kurreeman et al. [17]
		Multiple sclerosis	0.84 (0.80–0.88) ^α^	0.001	Wang et al. [18]
*IL2RB*	rs3218253	Rheumatoid arthritis	1.11 (1.05–1.17)	<0.001	Barton et al. [19]
*PDCD1*	rs2227981	Ankylosing spondylitis	1.25 (1.02–1.54)	0.033	Lee et al. [20]
		Type 1 diabetes	1.33 (1.07–1.66)	0.011	Lee et al. [20]
*PTPN11*	rs2301756	Ulcerative colitis	1.81 (1.11–2.97)	0.018	Narumi et al. [21]
*ZAP70*	rs13420683	Crohn’s disease	0.44 (0.26–0.76) ^α^	0.003	Bouzid et al. [22]

A literature overview of the selected SNPs and their associations with susceptibility to autoimmune diseases. The odds ratio (OR) was shown of the minor allele versus major allele as defined in the current study. If necessary, ORs were re-calculated by (1/OR). Abbreviations: CI: Confidence Interval. ^α^ ORs and 95% CIs were converted to enable comparison with the current study.

**Table 2 cancers-13-01370-t002:** Investigated SNPs.

Gene	Protein/Receptor	Variant	SNP ID	Assay ID	No. of WT	No. of HET	No. of HVAR	MAF ^α^	HWE *p*-Values
*GZMB*	Granzyme B	c.128C > A	rs8192917	C__2815152_20	85	32	2	15%	0.606
*HLAA*	HLA-A	del/TTTA	rs60131261	C__88752962_10	65	46	8	26%	0.971
*IFNG*	IFN-γ	c.874A > T	rs2430561	Assay by design	29	54	36	53%	0.330
		−1616T > C	rs2069705	C__15944115_20	58	47	14	32%	0.354
		367-895C > T	rs2069718	C__15799728_10	49	49	21	38%	0.162
*IL10*	IL-10	c.378 + 284G > T	rs3024493	C__15983657_10	82	34	3	17%	0.813
*IL2RA*	IL-2Rα /CD25	c.64 + 5006A > G	rs2104286	C__16095542_10	47	50	22	39%	0.187
*IL2RB*	IL-2Rβ	c.−34 + 1055G > A	rs3218253	C__27917605_10	54	57	8	31%	0.168
*PDCD1*	PD-1	804 C > T	rs2227981	C__57931286_20	38	53	28	46%	0.262
*PTPN11*	Shp-2	333-223A > G	rs2301756	C__2978562_20	97	22	0	9%	0.267
*ZAP70*	Zap-70	−21-4127C > A	rs13420683	C__1278468_10	66	47	6	25%	0.519

An overview of the investigated SNPs in patients included in this study (*N* = 119). Abbreviations: WT: wild types, HET: heterozygous variants; HVAR: homozygous variants; MAF: minor allele frequency; HWE: Hardy–Weinberg equilibrium. ^α^ Frequencies of the minor allele are shown as observed in the current study.

**Table 3 cancers-13-01370-t003:** Baseline patients’ characteristics.

Clinical Characteristics	No. of Patients (%) or Median (Range)
Number of patients	119 (100%)
Age (years)	63 (24–83)
Gender	
Male Female	76 (64%) 43 (36%)
Ethnicity	
Caucasian Black Unknown	116 (97%) 1 (1%) 2 (2%)
BRAF V600E/K mutation	
Yes No	40 (34%) 79 (66%)
Prior treatment	
No Yes ▪ BRAF/MEK-inhibitors ^γ^ ▪ Ipilimumab ▪ BRAF/MEK-inhibitors and anti-CTLA4 therapy ▪ TVEC	89 (75%) 30 (25%) 14 7 4 5
Treatment regime	
Pembrolizumab Nivolumab	56 (47%) 63 (53%)
ECOG performance status	
0 1 2 Unknown	82 (69%) 31 (26%) 1 (1%) 5 (4%)
LDH (U/L)	211 (118–1523)
CNS metastases	
Yes No Unknown	17 (14%) 60 (51%) 42 (35%)

Table showing the baseline patients’ characteristics of all patients in the study (*n* = 119). Abbreviations: TVEC: talimogene laherparepvec; ECOG: Eastern Cooperative Oncology Group; LDH: Serum lactate dehydrogenase; U: Units; L: Liter; CNS: central nervous system. ^γ^ i.e., vemurafenib and cobimetinib, dabrafenib and trametinib.

**Table 4 cancers-13-01370-t004:** Univariable and multivariable associations of clinical baseline characteristics and SNPs with OS.

Univariable Associations				
	Test Variables	HR (95% CI)	*p* Value	Bias Corrected 95% CI
**Clinical Baseline Characteristics**				
ECOG PS	1 + 2 vs. 0	3.625 (1.988–6.610)	<0.001	2.037–6.651
LDH	Continue (U/L)	1.004 (1.002–1.006)	<0.001	1.000–1.005
Age	≥65 years vs. <65 years	1.695 (0.931–3.086)	0.084	0.910–3.194
Prior treatment	Yes vs. no	1.631 (0.873–3.048)	0.125	
CNS metastasis	Yes vs. no	1.822 (0.729–4.559)	0.199	
Gender	Female vs. male	1.258 (0.690–2.295)	0.454	
BRAF V600E/K mutation	Present vs. absent	0.838 (0.438–1.603)	0.594	
**Variants: Dominant Model**				
PDCD1 804C > T	CT + TT vs. CC	2.027 (0.998–4.118)	0.051	1.011–4.516
GZMB c.128C > A	CA + AA vs. CC	1.811 (0.945–3.364)	0.060	0.994–3.351
IFNG c.874A > T	AT + TT vs. AA	0.653 (0.341–1.251)	0.199	
ZAP70 −21-4127C > A	CA + AA vs. CC	0.738 (0.403–1.348)	0.323	
PTPN11 333-233A > G	AG + GG vs. AA	1.440 (0.691–3.003)	0.330	
IFNG 367-895C > T	CT + TT vs. CC	0.783 (0.432–1.418)	0.420	
IFNG −1616T > C	TC + CC vs. TT	0.843 (0.466–1.539)	0.571	
IL2RA c.64 + 5006A > G	AG + GG vs. AA	0.845 (0.465–1.535)	0.579	
IL2RB −34 + 1055G > A	GA + AA vs. GG	0.875 (0.483–1.586)	0.661	
IL10 c.378 + 284G > T	GT + TT vs. GG	1.088 (0.577–2.054)	0.794	
HLAA c.del/TTTA	del + del/TTTA vs. insTTTA	1.126 (0.623–2.037)	0.964	
**Variants: Recessive Model**				
IFNG −1616T > C	CC vs. TT + TC	0.305 (0.074–1.265)	0.102	
PDCD1 804C > T	TT vs. CC + CT	1.556 (0.814–2.976)	0.181	
IFNG c.874A > T	TT vs. AA + AT	0.709 (0.358–1.404)	0.324	
IFNG 367-895C > T	TT vs. CC + CT	0.732 (0.309–1.736)	0.479	
IL2RA c.64 + 5006A > G	GG vs. AA + AG	1.034 (0.480–2.226)	0.932	
**Multivariable Associations**				
	**Test Variables**	**HR (95% CI)**	***p*** **Value**	**Bias Corrected 95% CI**
**Clinical Baseline Characteristics**				
ECOG PS	1 + 2 vs. 0	3.626 (1.941–6.774)	<0.001	1.784–6.946
LDH	Continue (U/L)	1.003 (1.002–1.005)	0.002	1.000–1.005
Age	≥65 years vs. <65 years	1.887 (1.002–3.554)	0.049	0.992–3.820
**Variants**				
PDCD1 804C > T	CT + TT vs. CC	2.366 (1.111–5.036)	0.026	1.039–6.246

Univariable and multivariable analyses of associations between SNPs and OS (*N* = 119). Bootstrap results are shown for results with *p* < 0.1, including the bias-corrected 95% confidence interval (CI). Clinical baseline characteristics and SNPs with *p* < 0.1 in univariable analysis were included in multivariable analysis. Table shows multivariable associations with *p* < 0.05. Abbreviations: ECOG PS: Eastern Cooperative Oncology Group Performance Status; LDH: Serum lactate dehydrogenase; CNS: central nervous system; Del: deletion.

## Data Availability

The data presented in this study are available in this article and Appendix A.

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
