# Peer review of "Germline Variation in PDCD1 Is Associated with Overall Survival in Patients with Metastatic Melanoma Treated with Anti-PD-1 Monotherapy"

_cancers, 2021, doi:10.3390/cancers13061370_

Round 1
Reviewer 1 Report
In the present study “Germline variation in PDCD1 is associated with overall survival in patients with metastatic melanoma treated with anti- PD-1 monotherapy”, the authors by genotyping the blood samples for 11 single-nucleotide polymorphisms (SNP) in 9 genes demonstrate that a common upstream germline variant in the gene encoding for PD-1, PDCD1 (804C>T; rs2227981), was associated with significantly worse overall survival in patients with metastatic melanoma treated with PD-1 immune checkpoint inhibitors (ICI). The study also reveals that the PDCD1 804C>T carriers had significant lower mRNA expression in several tissues and a decreased fraction of peripheral PD-1+ CD4+ T cells, indicating that PDCD1 804C>T may affect clinical benefit from PD-1 ICIs by decreasing transcriptional initiation and expression of PD-1 in T cells. In conclusion, the authors highlight the clinical need to understand how germline genetics affects immune responses during immunotherapy and suggest the feasibility for future genome-wide association studies in immuno-oncology.
In general, this manuscript is well-written and presents interesting insights regarding the role of germline genetics in response to immunotherapy in metastatic melanoma patients, adding more evidence in this field of research.
(lane 160) “Each essay consisted of two allele-specific minor groove binding”. Please correct the typo “essay”
I was wondering if any of these patients had major or life-threatening immune-related adverse events.
It would be very interesting to see immunohistochemical PD-1 expression in tumor infiltrating lymphocytes and subcutaneous adipose tissue; and/or PD-L1 immunoexpression in tumor or immune cells in these patients with heterozygous or homozygous variation of PDCD1 804C>T. It is interesting that expression quantitative trait loci analysis showed lower gene expression of PDCD1 in human tissue, in particular in subcutaneous adipose tissue.
Reviewer 2 Report
This manuscript reports an observation that SNP rs2227981 located in the upstream of PD1 gene is associated with overall survival of patients treated with PD1 antibody. Although only 11 SNPs was screened and a validation cohort was missing, the finding reported here is interesting and may still have the merit to publish.
A minor comment: the authors should use KM plot to show the survival difference.
Reviewer 3 Report
This is a nice, albeit limited report, looking at selected SNPs, and concentrating on a PDCD1 SNP that has been reported in several other papers tube associated with autoimmunity. If the authors could substantiate the genetic data with some functional/expression analysis that is more convincing than what is currently shown that would increase the quality and interest in this report.
Page 2 Line 92: I assume SNPs in non-coding regions can also affect the expression of encoded proteins and this is not made clear in this sentence
The exclusion of nonsynonymous SNPs is intriguing - if they were strongly associated with autoimmunity where they still excluded. It seems highly likely that many SNP, including synonymous SNPs, will have functional effects that are yet to be understood.
I would like the effect of the SNP amino acid change to be included in Table 2.
Page 7, Line 238 should be rewritten to improve clarity - perhaps indicate HR and CI with each parameter
The GZMB SNP had p value 0.06 in univariable analysis - so I am unclear where the 0.405 details in the results come from (Line250)
The analysis of PDCD1 genotype and OS/PFS does not mention whether the variants were homozygous or heterozygous-is this impactful to the data analysis and results
The data shown in Figure 1 are not convincing - with no statistics to compare the data - I would also like to see the patient matched longitudinal data i.e does the change in PD1 expression over time differ in WT vs PD1 variant patients - and there needs to be some separation with response? Was there any difference in lymphocytic infiltration in tumours related to PDCD1 genotype?
Author Response
Please see the attachement.
